# Nanocelluloses and Related Materials Applicable in Thermal Management of Electronic Devices: A Review

**DOI:** 10.3390/nano10030448

**Published:** 2020-03-02

**Authors:** Kimiyasu Sato, Yuichi Tominaga, Yusuke Imai

**Affiliations:** National Institute of Advanced Industrial Science and Technology (AIST), Anagahora 2266-98, Shimoshidami, Moriyama-ku, Nagoya 463-8560, Japan; tominaga.yuichi@aist.go.jp (Y.T.); y-imai@aist.go.jp (Y.I.)

**Keywords:** nanocelluloses, thermal management, thermal conductivity, nanocomposites

## Abstract

Owing to formidable advances in the electronics industry, efficient heat removal in electronic devices has been an urgent issue. For thermal management, electrically insulating materials that have higher thermal conductivities are desired. Recently, nanocelluloses (NCs) and related materials have been intensely studied because they possess outstanding properties and can be produced from renewable resources. This article gives an overview of NCs and related materials potentially applicable in thermal management. Thermal conduction in dielectric materials arises from phonons propagation. We discuss the behavior of phonons in NCs as well.

## 1. Introduction

After solid-state devices appeared and they replaced electron vacuum tubes, miniaturization has been the hallmark of microelectronics industry [1]. In the current era, there are over a billion transistors in a typical integrated circuit. The electronic components generate a significant amount of thermal energy. In around 2010, power dissipated from the micro-processors reached 100 W·cm^−2^, an order of magnitude higher than usual hot plates [2]. Shrinking the size and enhancing the computing capability of circuit devices have been achieved at the cost of increasing power generation in a smaller volume of space. With the rapid increase in power density in modern electronics, efficient heat removal has become an emerging demand for electronic products [3]. If the heat is not dissipated properly, it may lead to a malfunction or shortening of machine life. Since there has been an intense demand from public and industry for portable, flexible and high-performance electronic devices, the trend of shrinking size and escalating density will surely continue.

Heat transfer can occur through radiation, convection or conduction. Heat removal (passive cooling) in electronics is typically governed by conduction. The effectiveness of a heat spreading material is related to its thermal conductivity [4]. Thus, the key point of the thermal management is to use electrically insulating materials that have higher thermal conductivities [5]. Watts per meter-kelvin (W·m^−1^·K^−1^) is the commonly used unit for thermal conductivity in the related disciplines. As the heat spreading materials, polymer-based composite materials containing inorganic fillers have been studied quite actively [6,7,8,9,10,11]. In the composite materials, inorganic fillers are often dispersed in matrix polymers. Organic polymers have significant advantages such as mechanical flexibility, light weight, good processability, high electrical resistivity and affordability. However, their thermal conductivity is generally low, on the order of 0.1–1.0 W·m^−1^·K^−1^ [12]. The organic polymers are usually regarded as thermally insulating materials. The studies on the binary composite materials aim to compensate the inferior thermal conductivity of the polymers via inorganic fillers addition.

Heat conduction in metals is dominated by free electrons. In dielectric materials, heat conduction arises from lattice vibrations. By analogy with the photons of the electromagnetic field, the quanta of the lattice vibrational field are referred to as “phonons” [13,14,15]. In the vicinity of electronic components, metals are not suitable for heat dissipation because of concerns about short-circuiting. Accordingly, the subject of the present review settles on phonon-driven heat transfer.

Cellulose is the most abundant polysaccharide in nature. Cellulosic materials that have at least one dimension less than 100 nm are referred to generally as nanocelluloses (NCs). NCs are bio-based nanomaterials that are continuously photosynthesized and accumulated in plants. Due to their abundance, biodegradability and renewability, the novel forms of cellulose have been generating much activity in the materials science field [16,17,18,19,20,21,22]. NCs can be mainly divided into three classes. Herein, we have used the terms CNF, CNC and BNC. Cellulose nanofibers (CNFs), that possess heterogeneous network structures with widths less than 100 nm, can be prepared from wood pulp by mechanical disintegration treatments [23]. Since the mechanical processes are expensive, chemical or enzymatic pretreatments are often carried out to facilitate fibers separation. As shown in Figure 1, cellulose is a linear chain of ringed glucose molecules and has a flat ribbon-like conformation [24]. Within source fibrils, there are regions where the cellulose chains are arranged in a highly ordered (crystalline) structure and regions that are disordered (amorphous like). Since the amorphous regions are susceptible to acids, they can be removed by acid hydrolysis. Eventual residues of the acid treatments are known as cellulose nanocrystals (CNCs). CNCs consist of rod-like cellulose crystals with widths and lengths of 5–70 nm and between 100 nm and several micrometers, respectively. Bacterial nanocelluloses (BNCs) are formed by aerobic bacteria. BNCs are formed as polymers by biotechnological processes from low molecular-weight carbon sources. Very recently, NCs and nanocomposites containing NCs have garnered attention as potential heat spreading materials [25]. It has been accepted that heat transfer at the nanoscale can differ entirely from that at the macroscale [26]. By leverage from the specificity, NCs and related materials would be applicable in the thermal management of electronic devices. This review aims to provide an overview of NCs and related nanocomposites potentially applicable in thermal management. The authors attempted to review the theory that lies behind the thermally conductive nanomaterials as well.

## 2. Heat Conduction in Nanocelluloses

Recent theoretical and experimental research on nanomaterials has revealed interesting heat transfer phenomena at the nanoscale. As stated above, heat is mainly carried by phonons in dielectric materials. Thermal conductivity due to phonons (*κ*) can be roughly estimated by the following equation:
*κ* = (*C*_p_*ν l*)/3
where *C*_p_ is the specific heat capacity, *ν* is the phonon group velocity, and *l* is the mean free path of phonons in the material. The mean free path of phonons expresses how far phonons can travel before they are scattered by lattice imperfections, electrons, and other phonons. In bulky materials, internal scattering dominates heat conduction. As the size (characteristic length) shrinks, however, phonon scattering at the structure boundaries or interfaces is enhanced. In nonmetallic systems, the mean free paths at room temperature are from 1 to 100 nm. In nanostructures, their characteristic lengths are smaller or comparable to the intrinsic phonon mean free paths. Such an effect reduces the effective mean free path of phonons and the phenomenon is referred to as “classical size effect” [27,28]. As a typical instance of the classical size effect, we can adduce silicon nanowires. Silicon nanowires display a thermal conductivity which is much lower than that of bulk silicon. The enhanced boundary phonon scattering at the nanowire surfaces would be the main reason that leads to the lower thermal conductivity [29,30,31,32].

As for organic polymers, there exist papers reporting increased thermal conductivity ascribed to nano-sized structures. Recent experiments show that polymers can be eminent thermal conductors when the polymer chains are straight and aligned in crystalline fiber forms. Ultra-drawn polyethylene nanofibers were found to have thermal conductivity up to 100 W·m^−1^·K^−1^ [33,34,35,36] and the value is three orders of magnitude greater than that of their amorphous counterpart. Organic polymer chains exhibit another unique property. In polymer chains, thermal conductivity decreases when they transfer from one-dimensional chains to a three-dimensional crystal. The tendency is entirely opposite to the aforementioned classical size effect. That is caused by inter-chain interactions, which work as scattering sources for phonon transport [37]. Anharmonic interactions between chains can lead to increased phonon-phonon scattering, which lowers the thermal conductivity of individual chains. The effect is similar to a phenomenon found in graphene sheets [38]. It has been known that once graphene sheets are stacked in graphite structure, interlayer interactions quench the thermal conductivity of this system.

Preceding studies on papers and woods have reported that thermal conductivity of usual cellulosic fibers is in the range of 0.1–0.4 W·m^−1^·K^−1^ [39,40]. That is coherent with the knowledge that thermal conductivity of organic polymers is generally low. However, in 2007, it was reported that a CNF film impregnated with an epoxy resin has thermal conductivity greater than that of neat epoxy resin [41]. Subsequently, NCs have appeared promising as the heat spreading materials. There are several polymorphs of crystalline cellulose (I, II and III) [42,43]. Cellulose I is the crystalline cellulose that is naturally produced by various organisms and it is often referred to as “native” or “natural” cellulose. Celluloses II and III can be produced via artificial treatments from cellulose I. The native celluloses are composed of two different allomorphs, cellulose I_α_ (triclinic) and I_β_ (monoclinic) [44]. The celluloses from higher plants such as woods are dominant in I_β_. Crystallographic studies by Nishiyama and coworkers revealed that cellulose I_β_ possesses more compact crystalline structure and is thermodynamically more stable than cellulose I_α_ [45,46]. Polymers with more compact and less perturbed configurations should be desirable for phonon travelling [47]. Since the heat-spreading materials are utilized under harsh environments for long durations, more stable materials should be employed. Accordingly, cellulose I_β_ would be more significant for the present subject.

Figure 2 gives schemas of the cellulose I_β_ structure observed along two directions. Cellulose sheets composed of the flat ribbon-like chains are stacked in parallel. Because of technical difficulties, direct thermal conductivity measurement of cellulose I_β_ single crystals has not been reported to date. However, a prediction by molecular dynamics (MD) simulations was reported recently [48]. According to the MD simulations, the thermal conductivity in the chain direction was predicted to be around 5.7 W·m^−1^·K^−1^ at 300 K. When compared with the ultra-drawn polyethylene nanofibers, the value is not very high. But it is significantly higher than those of usual organic polymers. The thermal conductivity values in other directions are predicted to be less than 1 W·m^−1^·K^−1^. In the transverse direction, the cellulose sheets are stabilized by a network of hydrogen bonds [49,50]. In the stacking direction, the inter-sheet stability can be attributed to out-of-plane van der Waals interactions between the backbone ring structures [51]. The weak bondings or interactions would result in a lower thermal conductivity along the two directions. Thus, the inherent heat transfer capability of cellulose I_β_ relies on phonons traveling along the chain direction in its crystal structure.

Considering the aforementioned discussion, the superior thermal conductivity of NCs should stem chiefly from two factors: diminished amorphous regions and fewer frequency of inter-chain interactions. The two major factors are summarized in Figure 3. During the production process of NCs, the amorphous regions are removed by acid hydrolysis (Figure 3a). In amorphous polymers, random curvilinear polymer chains lead to structural scattering and phonons cannot propagate far [12]. In polymeric bodies composed of crystalline and amorphous regions, the higher degree/fraction of crystallinity leads to superior thermal conductivity [53,54]. The geometrical dimensions with larger specific surface area would increase ratio of polymer chains free from the inter-chain interactions (Figure 3b). As is stated above, inter-chain interactions work as scattering sources for phonon transport. In the isolated cellulose chains, the mean free path of phonons would become longer than that of bulky cellulose materials. The phonon scattering inhibition mechanisms described in Figure 3 should contribute to enhanced thermal conductivity of NCs.

## 3. Thermally Conductive Materials Containing Nanocelluloses

During the last decade, papers proposing novel heat-dissipation materials containing NCs have drastically increased in number. The related works published in scientific archival journals are summarized and assorted in Table 1. They are mainly divided into three classes based on the roles NCs are assuming; neat NCs as heat-dissipation materials; NCs as thermally conductive fillers; and NCs as scaffolds to maintain composite bodies. The three classes are illustrated in Figure 4. This section focuses on describing the features of the proposed heat dissipation materials.

The thermal conductivity values given are the highest reported in the reference papers. Thermal conductivity values vary depending on many factors such as solids loading, filler sizes, filler alignment, and so on. Comparing the given values naively is not very meaningful.

### 3.1. Neat Nanocelluloses

In 2015, Uetani and coworkers reported thermal conductivity measurements of non-woven sheets composed of various CNFs [55]. Among the evaluated specimens, in-plane thermal conductivity of a CNF sheet with the highest crystallinity reached 2.5 W·m^−1^·K^−1^. In 2017, the identical group reported an attempt to align cellulose chains in BNC hydrogels by drawing [56]. The BNC sheets exhibited anisotropy of thermal conductivity between the drawn and transverse directions. It was clarified that CNFs possess superior thermal conductivity ascribable to the cellulose chains and CNFs have been proved applicable for heat-spreading materials. Furthermore, directional control of the cellulose chains might lead to readily available heat guides.

### 3.2. Nanocelluloses as Heat-Conducting Fillers

As described in Section 2, Shimazaki and coworkers fabricated a CNF film embedded in an epoxy resin and evaluated its thermal conductivity [41]. The thermal conductivity value was 1.1 W·m^−1^·K^−1^, which is about 7 times higher than that of the pure epoxy resin. Following this, several groups have reported their attempts to utilize NCs as thermally conductive fillers [57,58,59]. The main purpose of employing matrix polymers would be to obtain optically transparent nanocomposites. Transparent polymeric composites functionalized by crystalline nanofillers are potentially useful in electronic and optical apparatuses [60]. Optical losses due to light scattering in composites are related to filler size and difference in refractive index between the fillers and the matrices [61,62]. The refractive index of cellulose is 1.5–1.6 in the optical wavelength range and is similar to those of transparent polymers [63]. When well-dispersed NCs are impregnated with transparent polymers, thermally conductive and optically transparent nanocomposites would become available. Actually, simultaneous pursuit of optical transparency and superior thermal conductivity has been reported in the preceding papers [41,58]. Chowdhury et al. emphasize another positive effect arising from the matrix polymers; filling up voids in the pristine NC materials with the polymers would reduce interfacial thermal resistance inside the system [59].

In composite materials, the phonon transports are hampered by scattering at filler–matrix interfaces. Thus, the thermal conductivity of polymers filled with NCs cannot surpass the intrinsic value of NCs (5.7 W·m^−1^·K^−1^). Added extra values such as optical transparency and a lightweight property might become the main indices for practical use.

### 3.3. Nanocelluloses as Composite Scaffolds

Owing to their high aspect ratio and extensive hydrogen bondings, films consist of an interconnected and entangled CNF network structure have an ability to sustain mechanical stress [64,65]. The films have remnant porosity from the gaps within the CNF network. Many attempts to enhance thermal conductivity by embedding inorganic crystallites in the pores have been reported. In the scheme, CNFs retain outer shapes and mechanical flexibility of the composite materials without impairing thermal conductivity of the inorganic crystallites heavily. To the authors’ knowledge, that can hardly be achieved with other organic polymers.

In 2014, Zhu and coworkers reported about the effect of hexagonal boron nitride (h-BN) nanosheets incorporated in CNF sheets [66]. The achieved thermal conductivity value was 145 W·m^−1^·K^−1^ for 50 mass% h-BN nanosheets. h-BN is a BN polymorph with a layered structure analogous to graphite. h-BN possesses wide band gap and can be treated as an electrical insulator [67,68]. Since h-BN boasts high thermal conductivity along the basal plane (400 W·m^−1^·K^−1^), it has been intensely studied as thermally conductive fillers [69,70,71,72]. High anisotropy in the crystal structure allows us to exfoliate h-BN crystallites and the resultant h-BN nanosheets also have drawn attention in the materials science discipline [73,74,75,76,77]. Even after exfoliation, h-BN nanosheets should retain the thermal conductivity along the basal plane. It has been accepted that the use of nanosheets should result in composite thermal conductivity that is either superior to that with particulate fillers or needs smaller solids loading to achieve the same performance [78,79,80,81]. When the nanosheet fillers are employed, frequency of filler-filler contact is increased and delocalized thermal conductive pathways form. More environmentally-aware works aiming similar scheme have been published recently. A method to prepare h-BN nanosheet/CNF composite films without utilizing organic solvents was reported [82]. A soy-derived protein was employed for surface modification of h-BN nanosheets to obtain h-BN nanosheets/CNF composites [83]. A work reported by Chen et al. should correspond to a derivative of the methodology. They attempted to fill up pores left in the h-BN nanosheets/CNF composite materials by epoxy impregnation [84].

The structural similarity between graphitic carbon and h-BN invoked studies on BN nanotubes (BNNTs) [85,86]. The thermal conductivity of BNNTs has been estimated to be several hundred W·m^−1^·K^−1^ [87,88] and thermal conductivity improvements of polymers via BNNT addition have been reported [89,90]. In 2017, Zeng et al. reported thermally conductive films composed of BNNTs and CNFs [91]. Nonetheless, it should be stated that BNNTs might be cytotoxic and must be utilized cautiously [92]. Long and fiber-shaped inorganic nanomaterials show acute cytotoxicity even though their low aspect ratio counterparts are harmless.

Graphene, a form of carbon that consists of a single layer of atoms arranged in a honeycomb lattice, exhibits quite high thermal conductivity (5000 W·m^−1^·K^−1^) [93]. It is possible to enhance the thermal conductivity of polymers by adding small quantity of graphene [94]. Song and coworkers have prepared CNF-based films containing graphene or graphene derivatives (graphene oxide/reduced graphene oxide), which possess superior thermal conductivity [95,96,97]. But it is obvious that graphene is highly electrically conductive. Electron mobility in graphene can exceed 15,000 cm^2^·V^−1^·s^−1^ at room temperature [98]. When electrically conductive fillers are employed, the electrical insulation property of the composite bodies should be considered. Guo et al. proposed a method to ensure electrical insulation of composites containing graphene fillers [99]. They arranged nano-sized Al_2_O_3_ particles between graphene fillers in ternary composites comprising of graphene, Al_2_O_3_ and CNF to avoid formation of electrical conduction paths. The resultant composites could be regarded as insulators.

Diamond is electrically insulating and boasts high thermal conductivity (2000 W·m^−1^·K^−1^). Nanodiamond (ND) is known as a member of the nanocarbons and has become a subject of active research [100,101,102]. Although NDs are not free from the classical size effect, NDs should inherit the superior property of bulk diamond to a degree [103]. In 2017, Song et al. reported that addition of ND particles to CNF films enhances thermal conductivity of the films efficiently [104]. Among the variety of NDs, ND materials prepared by detonation synthesis are distinguished from others by a moderate production cost and commercial availability. Sato and coworkers have prepared thermally conductive nanocomposites of CNF and detonation-produced ND, aiming prompt industrial applications [105,106]. Successively, Tominaga et al. reported their attempt to refine the CNF/ND composites by adding other inorganic fillers such as h-BN and α-Al_2_O_3_ [107].

Aluminum nitride (AlN) is known as a high thermal conductivity non-metallic solid [108]. Its intrinsic thermal conductivity has been estimated to be around 320 W·m^−1^·K^−1^ [109,110]. Very recently, attempts to prepare thermally conductive composites of CNF and nano-sized AlN particles have been reported [111,112]. Although AlN ceramics (sintered bodies) have assumed various significant roles in industries, AlN fine particles are difficult to handle [113]. The surface of AlN particles is chemically unstable. In the presence of moisture, AlN particles decompose forming aluminum hydroxide and ammonia [114,115]. Proper and efficient methods to protect the AlN particles’ surface should be employed, or the decomposition reaction of AlN fillers will cause device failures.

## 4. Conclusions and Perspectives

In this review, we have presented an overview of NCs as heat-spreading materials. Neat NCs can propagate thermal energy relatively well. Within the thermally conductive nanocomposites, NCs can fulfill roles both of the fillers and the continuous phases. Versatility would be a noteworthy feature of NCs. NC-based materials are also desirable from a sustainable development perspective. In the near term, process refinements would lower the production cost of NCs [116]. As a new class of NCs, ultrathin films of cellulose have been emerging [117]. The ultrathin films would be available for optoelectronic devices, sensors, and so on. NCs and the related materials are surely prime candidates applicable in the thermal management of electronic devices.

As far as the higher thermal conductivity remains paramount, combining NCs with inorganic fillers should be the most promising scheme. Indeed, as can be seen in Table 1, the inorganic filler-bearing specimens tend to exhibit superior thermal conductivity. Meanwhile, a high filler content leads to undesirable weight gain, cost rise and poor processability. Favorable materials should be developed for use considering their mechanical properties, optical properties, safety to living bodies, and so on.

As for NCs and related materials, theoretical works, that are essential for understanding nanoscale heat transfer, are scarce. Knowledge accumulated in the phonon engineering discipline might be of help [118]. Additional theoretical and experimental data should pave the way for applications of the NC-based materials in the electronics industry.

## Figures and Tables

**Figure 1 nanomaterials-10-00448-f001:**
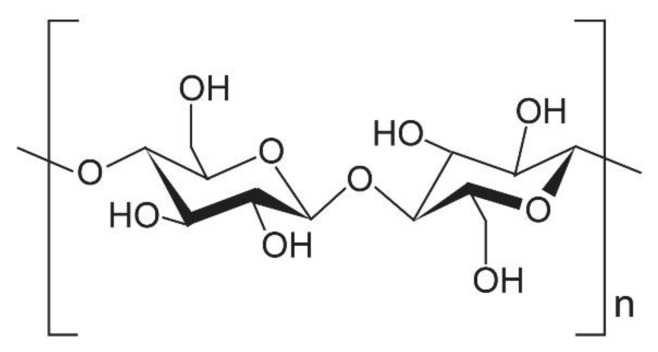
The repeating unit of cellulose.

**Figure 2 nanomaterials-10-00448-f002:**
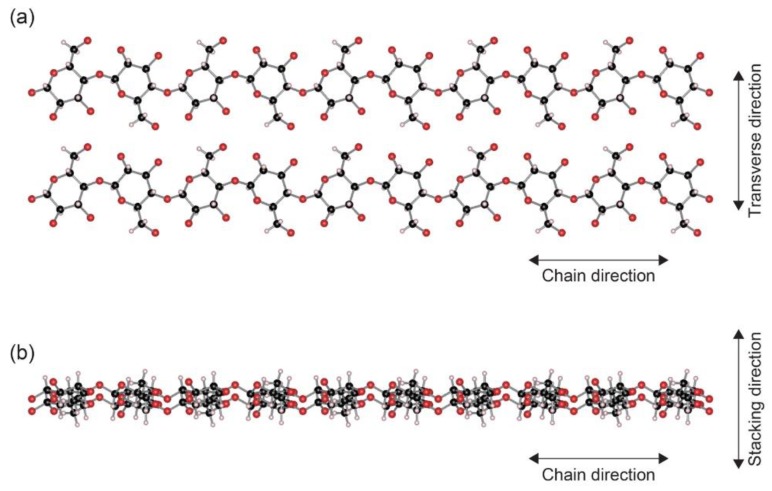
Cellulose I_β_ structure viewed along [001] (**a**) and [010] (**b**) directions. The schemas were produced based on the crystallographic parameters given in Nishiyama et al. [45], utilizing a visualization program *VESTA 3* [52]. Carbon, oxygen and hydrogen atoms are colored black, red and white, respectively.

**Figure 3 nanomaterials-10-00448-f003:**
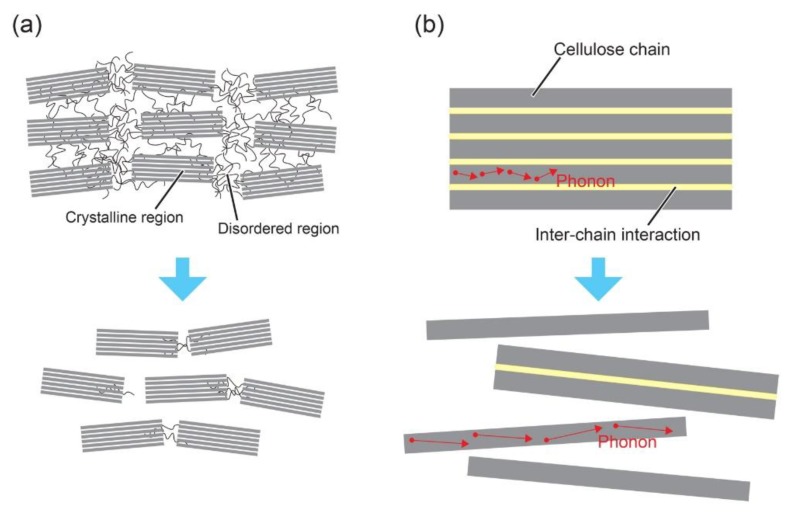
The two major factors that bring in the superior thermal conductivity of nanocelluloses (NCs). (**a**) Diminished amorphous regions; (**b**) Fewer frequency of inter-chain interactions.

**Figure 4 nanomaterials-10-00448-f004:**
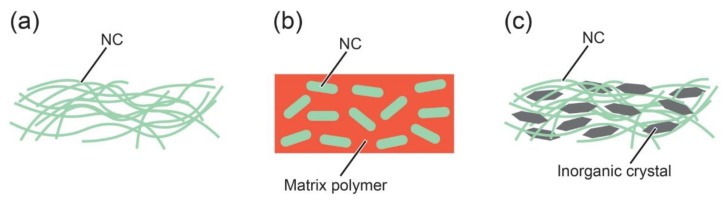
The roles NCs are assuming in the heat-dissipation materials. (**a**) Neat NCs as heat dissipation materials; (**b**) NCs as thermally conductive fillers; (**c**) NCs as scaffolds to maintain composite bodies.

**Table 1 nanomaterials-10-00448-t001:** NC-related heat dissipation materials proposed so far.

Class	NCs Utilized	Matrix Polymer	Inorganic Crystals Incorporated	Thermal Conductivity (W·m^−1^·K^−1^)	References
Neat NCs	CNF			2.5	[55]
	BNC			2.1	[56]
Fillers	CNF	Epoxy resin		1.1	[41]
	CNC	Polypropylene		0.4	[57]
	CNF	Acrylic resin		2.5	[58]
	CNC	Poly(vinyl alcohol)		3.5	[59]
Scaffolds	CNF		h-BN nanosheet	145	[66]
	CNF		h-BN nanosheet	23	[82]
	CNF		h-BN nanosheet	7	[83]
	CNF	Epoxy resin	h-BN nanosheet	3.1	[84]
	CNF		BNNT	21	[91]
	CNF		Graphene	12.6	[95–97]
	CNF		Al_2_O_3_, Graphene	8.3	[99]
	CNF		ND	11	[104]
	CNF		DND	4.8	[105,106]
	CNF		DND, Al_2_O_3_, h-BN	6.2	[107]
	CNF		AlN	5.1	[111,112]

h-BN = hexagonal boron nitride, BNNT = boron nitride nanotube, ND = nanodiamond, DND = nanodiamond produced by detonation.

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
