# Peer review of "Nanocelluloses and Related Materials Applicable in Thermal Management of Electronic Devices: A Review"

_nanomaterials, 2020, doi:10.3390/nano10030448_

Round 1
Reviewer 1 Report
This is a good review and I have only minor comments on it. The only critical comment is actually that cellulose IV is not considered an own polymorph but it has been revealed more than ten years ago that it is only a disordered form of cellulose I (see for example Biomacromolecules, https://pubs.acs.org/doi/pdf/10.1021/acs.biomac.9b01377)
Other comments:
Consider that also amorphous thin films (<100 nm) are considered cellulose nanomaterials. A good review on that can be found in Frontiers in Chemistry, https://www.frontiersin.org/articles/10.3389/fchem.2019.00488/full).
I would recommend to add that CNC and NFC can be obtained at different lengths, e.g. algal CNC can have a length of up to ten microns while wood based CNC is juts 100-300 nm. A short discussion on this point would be beneficial for the review. Maybe the authors can add a column on the dimensions of the used nanocellulose in Table 1.
Author Response
The authors very much appreciate the quite productive comments. We modified the manuscript according to the given comments.
We omitted the description about cellulose IV and added the paper concerning polymorphism of the celluloses to the References (Ref. [42]).
Since the ultrathin films of cellulose should be compatible with electronic devices, we added a description about the thin films in the “Conclusions and Perspectives” section. Further, we added the concerning review paper to the References (Ref. [116]).
Although the longer CNCs from algae have not been utilized for thermal dissipation, we are quite interested in the thermal properties of them. We would like to follow future developments.
Reviewer 2 Report
The paper is well prepared and highly actual by the topic, so it may be very attractive for the readers. I do not have any specific comment or suggestions to it; os it may be published as it is.
Author Response
The authors are so pleased to learn that the reviewer gave us the favorable comment.
Reviewer 3 Report
The topic of the paper is very interesting and timing. However the manuscript is not at the scientific level that is required for a publication in Nanomaterials. It is a summary of published information but it lacks of a critical discussion of the published state of the art.
The review paper has two main parts: one related to the nanocellulose (NC) and one related to the thermal management of electronic devices. The first part dealing with NC does not have an adequate scientific depth and does not show a high expertise on the production and application of NC to be able to critically examine the state-of-the-art and express informed views and provide guidance/ideas of future developments of the research topic. The second part shows a deeper understanding of the topic.
In summary, I cannot recomend the publication of the paper as it is now.
SPECIFIC COMMENTS:
In line 52 it is mention that "the nomemnclature of NCs has not been establish perfectly", which is not true since the ISO ISO/TS 20477:2017 has unified the terminologywith the NC nomenclature since 2017. In Line 53 "CNFs can be prepared by high-pressure homogenization with enzymatic pretreatments" seems like this is the main production process which is not true. The following lines can be also improved. It is suggested that authors are concise and refer to other reviews as for example in the Handbook of Nanomaterials for Industrial ApplicationsMicro and Nano Technologies. Chapter 5 - Nanocellulose for Industrial Use: Cellulose Nanofibers (CNF), Cellulose Nanocrystals (CNC), and Bacterial Cellulose (BC) 2018, Pages 74-126https://doi.org/10.1016/B978-0-12-813351-4.00005-5 The paper content no precise information with mistakes as in Line 145 since it is not true that amorphous regionsare removed by homogeneization or acid hydrolisis. This is only true in acid hydrolisys. The level of English is poor to be published Cost issues should be considered for the application.
Author Response
The central theme of the present manuscript is, as is stated in the main text, to foresee application of nano-sized celluloses in thermal management. We ascribed the superior thermal conductivity of CNCs to phonon travelling phenomena at nanoscales. Such the discussion has never appeared before in the disciplines related to CNCs. We appreciate if the reviewer accords the significance of the discussion.
Additionally, we modified the manuscript as follows:
The description "the nomenclature of NCs has not been establish perfectly" was deleted.
Round 2
Reviewer 3 Report
I recomend the authors to discuss point by point all the previous reviewer report and not only consider one of the coments.
Special attention should be given to wrong statements and confuse sentences as already mentioned.
Author Response
-In line 52, authors have removed "the nomenclature of NCs has not been establish perfectly", but they should have included the recommendation of reviewer related to the ISO ISO/TS 20477:2017 which unified the terminology with the NC nomenclature since 2017.
Line 52: We touched the International Standard (ISO/TS 20477:2017) and cited as Ref. [23].
-In Line 53 "CNFs can be prepared by high-pressure homogenization with enzymatic pretreatments" seems like this is the main production process which is not true. The following lines can be also improved. It was suggested that authors were concise and refer to other NC reviews.
Line 54: We amended the description on CNF preparation methods and cited a relevant review that has been published very recently, as Ref. [24].
-The paper content no precise information with mistakes as in Line 145 since it is not true that amorphous regions are removed by homogenization or acid hydrolysis. This is only true in acid hydrolysis.
Line 145: We deleted the words “high-pressure homogenization or”.